# FROM STEERING VECTORS TO CONCEPTORS AND BEYOND: COMPOSITIONAL AFFINE STEERING MECHANISMS FOR LLMS

## ABSTRACT

Controlling and understanding the representations of large language models (LLMs) remain central challenges as they become more powerful. In this paper, we combine conceptor theory with recent advances in activation steering to develop a novel framework that generalizes both approaches for provably optimal affine steering. Conceptors characterize sets of neural network activations, representable as ellipsoids, and they act as soft projection matrices, enabling precise and flexible control over LLM activations while offering deeper insights into their internal representations. Our framework derives optimal affine steering functions from first principles, outperforming traditional additive steering methods across in-context learning tasks. Additionally, we use a Boolean algebra over conceptor matrices that allows for the composition of multiple steering objectives. Empirical results demonstrate that this approach surpasses existing methods for combining steering vectors. By uniting conceptor theory with activation steering, this work provides not only a more powerful tool for controlling LLM outputs, but also a principled approach for better understanding the internal mechanisms governing model representations and behavior.

## 1 INTRODUCTION

Large language models (LLMs) have rapidly advanced AI capabilities (Xu & Poo, 2023), but their potential to spread misinformation (Pan et al., 2023), reinforce biases (Gallegos et al., 2024), and develop harmful behaviors (Shevlane et al., 2023) highlights the urgent need for methods to understand their internal workings and reliably control their outputs. Various methods, including reinforcement learning from human feedback (RLHF) (Ouyang et al., 2024), supervised fine-tuning (Devlin et al., 2019), and prompt engineering (Liu et al., 2023), have been proposed to steer LLM outputs toward desired patterns. However, RLHF and fine-tuning are computationally expensive and struggle with generalization (Bottou et al., 2018; Amodei et al., 2016), while prompt engineering often produces inconsistent results (Chen et al., 2023).

Activation steering (Turner et al., 2023; Li et al., 2023; Park et al., 2024; Subramani et al., 2022; Singh et al., 2024) has recently been proposed as a new steering method that works by directly modifying the model's activations at inference time without changing the model's parameters or relying on expensive optimization techniques. A steering vector that represents desired behavior can be computed directly (Turner et al., 2023; Subramani et al., 2022; Li et al., 2023) or contrastively from positive and negative examples (Rimsky et al., 2024b). It typically involves caching a set of token activation vectors from an LLM's forward pass on prompts that represent desired patterns (directly, *e.g.* "wedding", or contrastively, *e.g.* "love" - "hate"). These vectors are then subtracted or averaged to form a steering vector which can then be added onto a new forward pass to steer the model toward the desired pattern. This approach has shown to be effective at capturing and steering toward a wide range of patterns describing things like concepts (*e.g.,* weddings, love) (Turner et al., 2024), functions (*e.g.,* antonyms, synonyms) (Todd et al., 2024), and more complex behaviors (*e.g.,* truthfulness, power-seeking) (Rimsky et al., 2024a). However, the performance of activation addition is not always reliable (Turner et al., 2024; Price et al., 2024; Tan et al., 2024; Cao et al., 2024).

This paper introduces a more general framework for steering LLMs using activation engineering. Instead of averaging or subtracting a set of activation vectors to form a steering vector without much theoretical grounding, we derive an optimal linear and affine steering function and connect our results to existing work on conceptors (Jaeger, 2014b). Instead of manipulating the LLM's activations using vector addition, the activations are (softly) projected using a matrix-vector multiplication with the steering matrix, and optionally translated by an additional steering bias vector. We further present a Boolean algebra on linear steering matrices which allows arbitrary composition of our proposed linear steering functions. We apply our new linear and affine steering mechanisms to a set of tasks that are used in the literature on activation steering. We present results on function vectors (Todd et al., 2024) in Section 4.1, and show how these functions can be combined using the Boolean algebra on conceptors in Section 4.2.

## 1.1 RELATED WORK

Early work in activation steering explored the potential for modifying the internal activations of pre-trained language models (LLMs) to control their output at inference time, without requiring further training. Subramani et al. (2022) introduced the concept of "steering vectors", which, when added to the hidden states of a language model decoder, could steer the generation towards a target sentence with high accuracy. Their method involved optimizing a steering vector specific to each target sentence, achieving near-perfect BLEU scores on various English sentences. They also demonstrate the use of vector arithmetic for unsupervised style transfer. However, this approach's reliance on gradient descent for each sample limited its practical applicability to larger language models. Turner et al. (2023) proposed "Activation Addition," a more efficient method for calculating steering vectors by computing the difference in activations between prompt pairs designed to elicit contrasting behaviors. They demonstrated this technique's effectiveness in steering GPT-2-XL's output towards desired sentiments, topics, and styles, showcasing the potential for controlling LLMs without extensive computational overhead. Rimsky et al. (2024b) further built on this method to propose "Contrastive Activation Addition" where the steering vector would be formed using a dataset containing a large set of contrasting pairs instead of a single pair, as in Activation Addition. Li et al. (2023) independently developed "Inference-Time Intervention" (ITI), which utilizes linear probes to identify specific attention heads associated with truthful statements. By intervening on the activations of these heads, they were able to increase the model's truthfulness. Compared to activation addition, ITI focuses on causal interventions on specific components rather than a broader activation space modification.

Several follow-up papers have proposed improvements on the general activation addition methods. Wang et al. (2024) propose a method designed to improve the truthfulness of LLMs by adaptively adjusting the intensity of activation steering based on the truthfulness of the generated text. Stickland et al. (2024) fine-tune the LLM to minimize the KL-divergence between the model with the steering vector (as the student model) and the model without steering vector (as the teacher model) in order to mitigate detrimental effects of the steering vector on general model capabilities. Jorgensen et al. (2023b) focus on improving activation steering in language models by a technique called "mean-centring" for generating steering vectors, which aims to incorporate dataset-specific properties into the steering vectors. Various further papers have explored activation steering for different applications. Wang & Shu (2024) introduce "trojan steering vectors" to compromise the safety of LLMs. Rahn et al. (2024) aim at improving the performance of LLM agents in various tasks by learning a steering vector that encourages the agent to be more explorative. Qian et al. (2024) trace trustworthiness representations during training and find that steering vectors extracted from earlier pre-training checkpoints can be used to enhance the trustworthiness of models fine-tuned for specific tasks. Ghandeharioun et al. (2024) discover that activation steering is effective in bypassing safety filters. Price et al. (2024) were able to reduce the likelihood of backdoor behavior in LLMs using contrastive activation addition, but were unable to eliminate the vulnerability completely. Lu & Rimsky (2024) examines bias representations in Llama-2-Chat, and uses activation addition to steer the model's responses toward or away from stereotypes. Although preliminary results from activation addition are promising, the method's shortcomings have also been investigated. Tan et al. (2024) show that while steering can work well in the right circumstances, there remain many technical difficulties of applying steering vectors to guide models' behaviour at scale. Cao et al. (2024) argue that existing methods for extracting steering vectors, which rely on directly calculating activation differences from contrastive prompt pairs, can lead to suboptimal results, particularly in

alignment-related scenarios. They propose optimizing the steering vectors to directly influence the generation probability of contrastive human preference data pairs.

Most activation addition work has been empirical, but more theoretically grounded approaches to activation steering are now emerging. Todd et al. (2024) introduce the concept of "function vectors" (FVs) in large language models (LLMs), which represent specific input-output mappings within the model's activation space. They argue that these FVs are distinct from simple semantic vector offsets and play a crucial role in the model's ability to perform in-context learning (ICL). Park et al. (2024) examine the Linear Representation Hypothesis, which posits that meaningful information in LLMs is encoded in linear subspaces within the activation space. The authors argue for the importance of the inner product as a key operation for understanding and manipulating these representations. The paper aims to clarify the theoretical foundations of techniques like activation steering, and highlights the importance of linearity in understanding how models represent and process information. Singh et al. (2024) present a theoretical framework for understanding affine steering functions and derive two optimal functions under different constraints. Interestingly, under their constraints of "guardedness", they show that the optimal affine steering mechanism is simple additive steering - which provides theoretical justification for existing steering approaches. They empirically validate the effectiveness of their proposed steering methods in mitigating bias and reducing toxicity.

It becomes clear from previous work that steering vectors alone are not expressive enough to reliably steer model behavior. However, the evidence for the linear subspace representation (Park et al., 2024) suggests that affine or linear methods should be sufficient to intervene on hidden representations with the effect of steering them towards certain behaviors. Furthermore, empirical evidence from the literature on concept erasure has shown that affine interventions are effective even for deep and nonlinear models (Ravfogel et al., 2022; Belrose et al., 2023).

## 2 A THEORETICAL FRAMEWORK FOR ACTIVATION STEERING

### 2.1 PRELIMINARIES

We follow the formalism for steering functions that was introduced by Singh et al. (2024). Let $\Sigma$ be an alphabet, *i.e.,* a finite and non-empty set. A language model $p$ is a distribution over $\Sigma^*$, the set of all strings over the alphabet $\Sigma$.

We further introduce $\mathcal{C}$ as the set of concepts that may be active in the current text sequence $s \in \Sigma^*$. These concepts may correspond to functions as in Todd et al. (2024), binary concepts as in Singh et al. (2024), or other, more complex, behaviors exhibited by language models.

Given a language model $m$, we define the following conditional distribution:

$$m_c(s) := m(s \mid C = c) \propto m(s)\mathbf{1}\{\phi(s) = c\}, \tag{1}$$

which expresses the probability of sampling a string $s$ with concept $c$ present.

Let $\texttt{enc} : \Sigma^* \to \mathbb{R}^D$ be a language encoder, a deterministic function from the set of strings to real-valued vectors. This need not be a specialized module – it could be, for example, the hidden activations of a decoder-only transformer model. With a fixed encoder function, we can define the following $\mathbb{R}^D$ random variable:

$$\mathbf{H}(s) = \texttt{enc}(s) : \Sigma^* \to \mathbb{R}^D, \tag{2}$$

which is distributed according to:

$$\mathbb{P}(\mathbf{H} = \mathbf{h} \mid C = c) = \mathbb{P}(\mathbf{H}^{-1}(\mathbf{h}) \mid C = c) = \sum_{s \in \Sigma^*} m_c(s)\mathbf{1}\{\mathbf{h} = \texttt{enc}(s)\} \tag{3}$$

We assume that $\mathbf{H}$ is of finite first and second moment and denote the concept-conditional means of $\mathbf{H}$ with respect to $c$ as $\mu_c$, the concept-conditional second moment as $\tilde{\Sigma}_c$, and the concept-conditional covariance matrix as $\Sigma_c$, all defined below:

$$\mu_c = \mathbb{E}[\mathbf{H}_c], \quad \tilde{\Sigma}_c = \mathbb{E}[\mathbf{H}_c\mathbf{H}_c^\top], \quad \Sigma_c = \mathbb{E}[\mathbf{H}_c\mathbf{H}_c^\top] - \mu_c\mu_c^\top \tag{4}$$

for all concepts $c \in \mathcal{C}$.

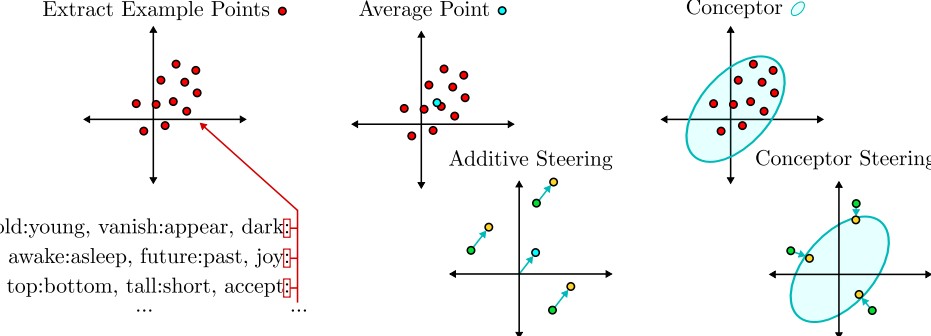

Figure 1: Illustration showing the basic geometric difference between additive and conceptor steering using a set of activations for the antonym task.

We are interested in functions that map representation-valued random variables to other representation-valued random variables. Such functions are called *intervention functions* (Singh et al., 2024). We are specifically interested in *steering functions* $f_c$, which are intervention functions that steer a given representation towards some concept $c$.

**Definition 1** ($\phi$-assisted steering function). *We define a steering function $f_c$ to be $\phi$-assisted, if it is of the form:*

$$f_c(\mathbf{H}(s)) = \begin{cases} f'_c(\mathbf{H}(s)) & \text{if } \phi(s) = c' \\ \mathbf{H}(s) & \text{if } \phi(s) = c, \end{cases} \tag{5}$$

*where $f'_c : \mathbb{R}^D \mapsto \mathbb{R}^D$ is a steering function and $\phi : \Sigma^* \mapsto \mathcal{C}$ is a concept encoding function.*

Singh et al. (2024) investigate such $\phi$-assisted steering functions. For the present paper, we instead consider *unassisted steering functions* which do not explicitly make use of a concept encoding function $\phi$ when steering the model at inference time, following prior work on activation steering Turner et al. (2023); Li et al. (2023); Subramani et al. (2022).

### 2.2 ADDITIVE STEERING FUNCTIONS

Additive steering functions have been the dominant approach to steering model behavior (Turner et al., 2023; Rimsky et al., 2024b; van der Weij et al., 2024).

**Definition 2** (additive steering function). *We define a function $f_c$ to be an additive steering function if it is of the form:*

$$f_c(\mathbf{H}(s)) = b_c + \mathbf{H}(s) \tag{6}$$

*where $b_c \in \mathbb{R}^D$ is the steering vector that corresponds to concept c.*

Typically, this additive steering vector is chosen to be $b_c = \mu_c$ where $\mu_c$ is, as defined above, the concept-conditional representation mean (Turner et al., 2023). In contrastive activation addition, the steering vector is chosen to be $b_c = \mu_c - \mu_{c'}$ where $c$ is the target concept and $c'$ is a contrastive concept. Recent work by Singh et al. (2024) has shown that, when guardedness is required (see Section 2.3), the optimal affine steering method for binary concepts simplifies to contrastive additive steering.

### 2.3 LINEAR STEERING FUNCTIONS

In the following, we will remove the constraint of guardedness that makes purely additive steering optimal (Singh et al., 2024), as we are interest purely in steering the behavior of a language model, and not in corresponding debiasing or concept erasure. As our goal is to find a more performant steering mechanism, we are now looking at the class of linear steering functions which we hypothesize are more expressive than additive steering functions.

**Definition 3** (linear steering function). *We define a function $f_c$ to be a linear steering function if it is of the form:*

$$f_c(\mathbf{H}(s)) = C\mathbf{H}(s) \tag{7}$$

*where $C \in \mathbb{R}^{D \times D}$ is the steering matrix that corresponds to concept c.*

As we do not want to rely on the concept function $\phi$ to apply our steering function, we instead rely only on the concept-conditional covariance matrix $\Sigma_c$. We now derive the optimal linear steering function that minimally changes the representation. Singh et al. (2024) derive an optimal affine steering function subject to the constraint that it also guards the representations against the concept that is being steered. We find that this is not an essential requirement for our purposes. Instead, we follow the approach by Jaeger (2014b) to define a *"conceptor" steering matrix* $C$ through an objective function whose first component pushes $C$ to act as a projection matrix for states that exhibit the target concept $c$, and whose second component adjusts how many leading directions of the covariance matrix $\Sigma_c$ should be effective for this projection.

**Definition 4** (optimal linear steering function). *We define the optimal linear steering function to be the function $f_c(\mathbf{H}(s)) = C\mathbf{H}(s)$ where $C$ is the conceptor matrix which solves the following optimization problem:*

$$C(\tilde{\Sigma}_c, \alpha) = \arg \min_C \mathbb{E}\left[\|\mathbf{H}_c - C\mathbf{H}_c\|_2^2\right] + \alpha^{-2}\|C\|_F^2 \tag{8}$$

*where $h$ is the representation, $\|\cdot\|_F$ is the Frobenius norm, and $\alpha$ is the regularization parameter also referred to as the conceptor's aperture.*

The aperture parameter $\alpha$ balances the trade-off between accurately representing the activation pattern and maintaining a generalized representation. This parameter allows us to tune how much of the concept's signal variance is captured or filtered by the conceptor. This allows a formal investigation into steering more general concepts beyond binary concepts.

When $\alpha$ is large, the eigenvalues $\mu_i$ approach 1 and $C$ approaches the identity matrix, causing the conceptor to allow for more signal components to pass through the projection of the representations with the conceptor matrix. Conversely, when $\alpha$ is small, the eigenvalues $\mu_i$ approach 0, causing the conceptor to allow for less variability. In the extreme case of $\alpha = 0$, the conceptor collapses to the zero mapping. This minimization problem uniquely specifies the conceptor $C(\tilde{\Sigma}_c, \alpha)$, and can be computed in closed form from $\tilde{\Sigma}_c$ and $\alpha$.

**Proposition 1.** *Let $\tilde{\Sigma}_c$ be the concept-conditional second moment of the random variable $\mathbf{H}(s)$ and $\alpha \in (0, \infty)$. Then, the conceptor $C(\tilde{\Sigma}_c, \alpha)$ is uniquely defined and can be directly computed as:*

$$C(\tilde{\Sigma}_c, \alpha) = \tilde{\Sigma}_c \left(\tilde{\Sigma}_c + \alpha^{-2}I\right)^{-1} \tag{9}$$

*The matrix $C(\tilde{\Sigma}_c, \alpha)$ is positive semi-definite with eigenvalues in the range $[0, 1)$.*

*Proof.* See Jaeger (2014b).

Where the context is apparent, we drop the function notation and denote the conceptor matrix simply by $C$. The conceptor matrix $C$ captures the principal directions and variances of a set of neural activation vectors. This structure can be visualized as a high-dimensional ellipsoid that describes the overall shape and spread of the activations' "underlying pattern" or state space region, see Figure 4.

Because conceptors are computed from the cloud of activation vectors and encode the correlations between activations, we expect that conceptors will be able to better capture the activation space of complex patterns compared to additive methods, which discard information about correlations.

### 2.3.1 COMBINING LINEAR STEERING FUNCTIONS WITH BOOLEAN OPERATIONS

We can combine multiple steering matrices using the Boolean operations on conceptors, as defined by Jaeger (2014b). These operations allow us to merge conceptors computed on different data samples to construct more complex steering targets. We begin by defining the OR operation on two conceptors, which is computed by summing the covariance matrices on which they are based. This operation can be understood as merging the data from which each conceptor was derived. The resulting conceptor is then computed based on the sum of these covariance matrices.

**Definition 5** (OR Operation on Conceptors). *Let $C_1$ and $C_2$ be two conceptors computed from covariance matrices $\Sigma_{c_1}$ and $\Sigma_{c_2}$, respectively. The OR operation, $C_1 \vee C_2$, combines these conceptors by adding their covariance matrices and is given by:*

$$C_1 \vee C_2 = (\Sigma_{c_1} + \Sigma_{c_2})\left(\Sigma_{c_1} + \Sigma_{c_2} + \alpha^{-2}I\right)^{-1}$$

*Using Equation 9, this can be rewritten as:*

$$C_1 \vee C_2 = \left( I + \left( C_1 (I - C_1)^{-1} + C_2 (I - C_2)^{-1} \right)^{-1} \right)^{-1}$$

Next, we define the NOT operation. This operation inverts the covariance matrix, producing a conceptor that captures data that co-varies inversely to the original conceptor.

**Definition 6** (NOT Operation on Conceptors). *Let $C$ be a conceptor derived from covariance matrix $\Sigma_c$. The NOT operation on a conceptor, denoted by $\neg C$, is computed by inverting the covariance matrix. The NOT operation is defined as:*

$$\neg C = \Sigma_c^{-1} (\Sigma_c^{-1} + \alpha^{-2} I)^{-1}$$

*Using Equation 9, this can be rewritten as:*

$$\neg C = I - C$$

Using the NOT and OR operations, we can now define the AND operation using de Morgan's law. The AND operation captures the intersection between the two conceptors.

**Definition 7** (AND Operation on Conceptors). *Let $C_1$ and $C_2$ be two conceptors. The AND operation, denoted by $C_1 \wedge C_2$, can be obtained using de Morgan's law: $C_1 \wedge C_2 = \neg(\neg C_1 \vee \neg C_2)$. This leads to the following formulation:*

$$C_1 \wedge C_2 = (\Sigma_{c_1}^{-1} + \Sigma_{c_2}^{-1})^{-1} \left( (\Sigma_{c_1}^{-1} + \Sigma_{c_2}^{-1})^{-1} + \alpha^{-2} I \right)^{-1}$$

*Using Equation 9, this can be rewritten as:*

$$C_1 \wedge C_2 = (C_1^{-1} + C_2^{-1} + I)^{-1}$$

These Boolean operations can be used to combine multiple conceptor steering matrices into more complex steering targets. Similar operations have been proposed for additive steering methods. Todd et al. (2024) propose a task arithmetic on function vectors and demonstrate it on a some toy tasks, while Subramani et al. (2022) use a vector arithmetic on steering vectors. The negation of additive steering vectors has been used widely in contrastive steering as introduced by Rimsky et al. (2024b). We note that the AND and OR operations on conceptor steering matrices do not clearly correspond to the addition operation on steering vectors. In Section 4.2, we compare combinations of steering vectors against combinations of conceptor-based steering matrices.

## 2.4 AFFINE STEERING FUNCTIONS

We now turn to the class of affine steering functions, in order to generalize the results on conceptors (Jaeger, 2014b), affine steering functions (Singh et al., 2024), and additive steering functions (Turner et al., 2023; Subramani et al., 2022; Li et al., 2023) into a more general framework of affine activation engineering.

**Definition 8** (affine steering function). *We define a function $f_c$ to be an affine steering function if it is of the form:*

$$f_c(\mathbf{H}(s)) = C\mathbf{H}(s) + b \tag{10}$$

*where $C \in \mathbb{R}^{D \times D}$ is the steering matrix, and $b \in \mathbb{R}^D$ is the steering vector, both of which corresponding to concept c.*

We define the *optimal affine steering function* in an analogous way to how we defined the optimal linear steering function, as the solution to an optimization problem.

**Definition 9** (optimal affine steering function). *We define the optimal affine steering function to be the function $f_c(\mathbf{H}(s)) = C\mathbf{H}(s) + b$ which solves the following optimization problem:*

$$\min_{C \in \mathbb{R}^{D \times D}, b \in \mathbb{R}^D} \mathbb{E} \left[ \|\mathbf{H}_c - C\mathbf{H}_c - b\|_2^2 \right] + \alpha^{-2} \|C\|_F^2 \tag{11}$$

In the following proposition, we derive the unique solution for the optimal affine steering function.

**Proposition 2.** *Let $\Sigma_c$ be the concept-conditional covariance matrix of $\mathbf{H}(s)$, $\mu_c$ its concept-conditional mean, and $\alpha \in (0, \infty)$. Then, the optimal affine steering function $f_c$, as defined above, can be directly computed as:*

$$C(\Sigma_c, \alpha) = \Sigma_c(\Sigma_c + 2\alpha^{-2}I)^{-1} \tag{12}$$
$$b(\Sigma_c, \alpha) = \mu_c - C(\Sigma_c, \alpha)\mu_c \tag{13}$$

*such that the final steering function is of the form:*

$$f_c(\mathbf{H}(s)) = Cx + b \tag{14}$$
$$f_c(\mathbf{H}(s)) = C(x - \mu_c) + \mu_c \tag{15}$$

*where $C = C(\Sigma_c, \alpha)$ and $b = b(\Sigma_c, \alpha)$.*

*Proof.* See Appendix A.1.

In this resulting steering mechanism, we can see connections to existing work. Jorgensen et al. (2023b) argue that mean centering the activations before applying the steering vector could improve performance – a similar operation is applied in our optimal affine steering mechanism.

## 3 METHODS

Given a finite sample $H \in \mathbb{R}^{D \times n}$ of $n$ representations from $\mathbf{H}_c$, we can approximate the covariance matrix with $\hat{\Sigma}_c = HH^T/n$. We empirically found that the conceptor-based steering method works best if it acts additively on the residual stream of a large language model (Elhage et al., 2021). If we consider the activation of the residual stream at layer $\ell$ right before the multi-head attention to be $h_\ell$, and the conceptor $C_\ell^c$ is computed using the covariance matrix on a sample of activation vectors exhibiting concept $c$ at layer $\ell$ becomes:

$$f_c(h_\ell) = h_\ell + \beta_c C_\ell h_\ell = (\beta_c C_\ell + I)h_\ell \tag{16}$$

where $\beta_c > 0$ is a hyperparameter. We can think of this as a "soft projection". A projection matrix has eigenvalues that are either zero or unity, but the conceptor matrix has "soft" eigenvalues between zero and unity. Thus, the conceptor "softly projects" the activation vector $h_\ell$ toward the pattern represented by $C_\ell$ by scaling its components according to the patterns' principal directions.

The setup for affine conceptor-based steering is analogous to the linear case but with the conceptor $C$ being computed as described in Equation 12 and the bias $b$ being computed as described in Equation 13. The activations $h_\ell$ are then steered with:

$$f_c(h_\ell) = h_\ell + \beta_c(C_\ell h_\ell + b) = h_\ell + \beta_c(C_\ell(h_\ell - \mu_c) + \mu_c) \tag{17}$$

where $\mu_c$ is the concept-conditional mean computed on the sample $H \in \mathbb{R}^{D \times n}$. We can think of this operation as a soft projection on mean-centered data, similar to what was proposed by Jorgensen et al. (2023b).

## 4 EXPERIMENTS

For our experiments, we will use EleutherAI's GPT-J 6B, GPT-NeoX 20B, and GPT-2 Small models, as done in previous works (Todd et al., 2024; Jorgensen et al., 2023a). For all experiments, we find optimal hyperparameters for each steering method at every layer. The details of our grid search for $\alpha$ and $\beta_c$ for conceptor-based steering and $\beta_{\text{add}}$ for additive steering can be found in Appendix A.3.2.

### 4.1 FUNCTION STEERING

We compare conceptor-based and additive steering mechanisms on their ability to steer a given model towards correctly executing a set of functions. We test both methods on GPT-J with 6B parameters and GPT-NeoX with 20B parameters. For each function, the described experiment will be repeated 5 times with different random seeds, and all reported results are averaged across across these five runs. The examples of the input-output functions come from the dataset by Todd et al. (2024). We use the following subset of five functions (Jorgensen et al., 2023a): antonyms

(e.g. good→bad), present-past (e.g. go→went), English-French (e.g. hello→bonjour), singular-plural (e.g. mouse→mice), country-capital (e.g. Netherlands→Amsterdam), and capitalize (e.g. word→Word). To ensure comparability of our results, we follow the work by Todd et al. (2024) as closely as possible. For more details, see Appendix A.3.1.

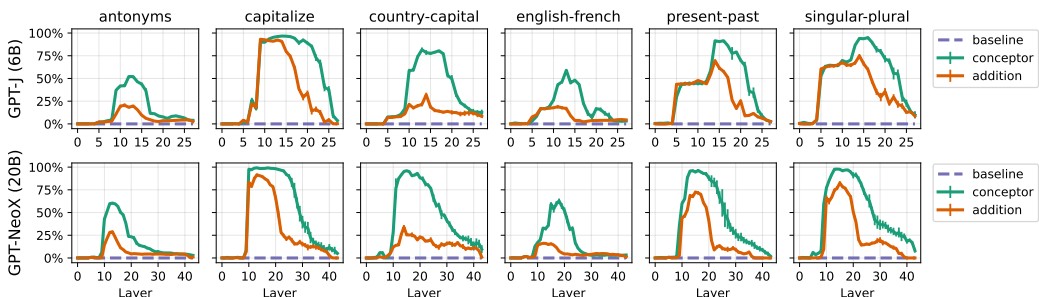

Figure 2: Comparison of the accuracy on all six function tasks for conceptor-based steering against additive steering across all layers for GPT-J and GPT-NeoX. For explanation, see main text.

The results in Figure 2 show that conceptor-based steering outperforms additive steering (the baseline method reported by Todd et al. (2024)) for every task on both tested models. Results show the best-performing model across a range of hyperparameters. It is clearly evident that conceptor steering is strictly more performant than additive steering across all tasks for most layers. Results for the complete hyperparameter sweep are presented in Appendix A.5. In line with previous findings (Todd et al., 2024; Jorgensen et al., 2023a), steering is most effective across layers 9-16 for GPT-J and layers 10-30 for GPT-NeoX.

Table 1: A comparison of affine conceptors, linear conceptors, activation vectors and mean-centered (MC) activation vectors on the GPT-J (6B) model, across simple function vector tasks. Results show the best performance across all hyperparameters and across all layers.

|  | antonyms | capitalize | country-capital | english-french | present-past |
|---|---|---|---|---|---|
| Addition | 20.54% | 93.16% | 32.04% | 18.88% | 69.66% |
| Addition (MC) | 31.20% | 95.00% | 63.90% | 34.32% | 83.32% |
| Linear conceptor | 52.14% | **96.68%** | 81.62% | 59.02% | 91.56% |
| Affine conceptor | **52.82%** | 96.26% | **85.32%** | **61.32%** | **91.88%** |

We also present results for affine conceptors that include a mean-centering operation as defined in Equation 17 and Section 3. The experiment is described in full detail in Appendix A.4. The results are shown in Table 1. The mean-centering improvement on additive steering, proposed by Jorgensen et al. (2023b) yielded a relative improvement over additive steering of as much as 99% on the country-capital task. Analogously, affine conceptors improved steering accuracy on some of the tasks, but the relative improvement was limited to no more than 5% in accuracy.

## 4.2 STEERING COMPOSITE FUNCTIONS

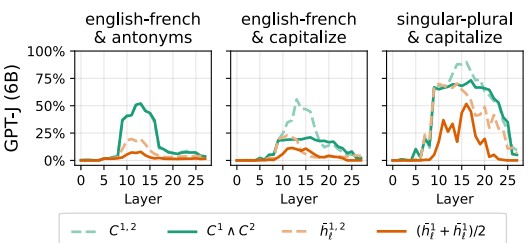

We further conducted experiments where two conceptors representing three different compound functions were combined using the AND operator. The input-output example dataset for this function was generated using GPT-4o and will be made available for the camera-ready paper. To present the baseline for how well non-combined steering mechanisms perform, we show results for the conceptor $C^{1,2}$ and the steering vector $h_\ell^{\bar{1},2}$ that were each computed on

Figure 3: Performance of additive steering and conceptor steering on composite functions.

the compound function directly. We then combine the conceptors computed on the individual functions $C^1$ and $C^2$ using the AND operation as $C^1 \wedge C^2$, and we combine the steering vectors $\bar{h}_\ell^1$ and $\bar{h}_\ell^2$ using their arithmetic mean $\frac{1}{2}(\bar{h}_\ell^1 + \bar{h}_\ell^2)$. Figure 3 shows the performance of all compared methods across all layers of the GPT-J model. In line with the results from Section 4.1, the conceptor baseline outperformed the additive baseline on all three tasks. The AND-combined conceptor outperformed the mean-combined steering vectors. On one of the three tasks, english-french & antonyms, the AND-combined conceptor even outperforms the additive baseline.

## 5    CONCLUSION

The integration of conceptor theory with activation steering provides a new lens through which to understand and manipulate large language models (LLMs). By deriving an optimal affine steering function from first principles, we establish a rigorous foundation for steering, addressing the limitations of existing additive methods. Conceptors, represented as ellipsoids, enable more precise control by capturing the full covariance structure of neural activations, which allows them to generalize beyond the simple vector offsets commonly used in additive steering. Moreover, the projection-based steering is inherently adaptive without an additional mechanism such as the one proposed by Wang et al. (2024), since activations residing within the conceptor's region would experience minimal change whereas activations outside of the conceptor's region experience a more substatial shift. The ability to project activation vectors softly through conceptors reveals how concepts are encoded and how they can be influenced without requiring model retraining. This positions conceptor-based steering not only as a tool for output manipulation but also as a method for interrogating and interpreting model behavior.

Additionally, the compositional nature of conceptor operations, implemented through Boolean algebra, offers a powerful mechanism for multi-task steering. By combining conceptors using operations like AND and OR, we are able to create composite steering objectives that outperform traditional methods of combining steering vectors. This demonstrates the versatility of our approach, allowing for more sophisticated control of LLMs, especially in multi-task scenarios where steering objectives may conflict or overlap.

In our experiments we show that conceptor-based steering outperformed addition-based methods across functions and combined functions. Despite its strengths, conceptor-based steering introduces additional complexity and computational cost. The need to compute covariance matrices, and the tuning of hyperparameters like aperture, increases the overhead compared to simpler additive methods. However, these trade-offs are justified by the gains in precision and control, especially in tasks where additive steering has proven insufficient. We mention also that the conceptor matrix can be fused with the attention head weights to not impact model latency.

While our framework demonstrates success across a range of tasks, further exploration is needed to understand its scalability to larger models and more diverse tasks. Investigating how conceptors interact with even more complex behaviors in LLMs, such as multi-turn dialogue or long-term reasoning, could provide further insights into the flexibility of this approach.

Our work unites conceptor theory and activation steering, offering a robust framework for both controlling and understanding LLMs. By deriving a provably optimal affine steering mechanism and introducing composable Boolean operations, we provide a method that not only surpasses traditional steering approaches but also lays the groundwork for more advanced activation engineering techniques. While challenges remain, the combination of theoretical rigor and empirical success positions conceptor-based steering as a powerful tool for the future of LLM control and interpretability.

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

# A  APPENDIX

## A.1  PROOF OF PROPOSITION 2

We aim to minimize the cost function defined in Equation 11:

$$L(C, b) = \mathbb{E}\left[\|\mathbf{H}_c - C\mathbf{H}_c - b\|_2^2\right] + \alpha^{-2}\|C\|_F^2 \tag{18}$$

The first derivative of this function is:

$$\frac{\partial}{\partial C}L(C, b) = \frac{\partial}{\partial C}\left(\mathbb{E}\left[\|x - Cx - b\|_2^2\right] + \alpha^{-2}\|C\|_F^2\right) \tag{19}$$

$$= -\mathbb{E}\left[(x - Cx - b)x^\top\right] + \alpha^{-2}C^\top C \tag{20}$$

$$= -\mathbb{E}\left[xx^\top - Cxx^\top - bx^\top\right] + \alpha^{-2}C^\top C \tag{21}$$

$$= -\tilde{\Sigma}_c + C\tilde{\Sigma}_c + b\mu_c^\top + 2\alpha^{-2}C \tag{22}$$

$$\frac{\partial}{\partial b}L(C, b) = \frac{\partial}{\partial C}\left(\mathbb{E}\left[\|x - Cx - b\|_2^2\right] + \alpha^{-2}\|C\|_F^2\right) \tag{23}$$

$$= -\mu_c - C\mu_c + b \tag{24}$$

The second derivative of this function is:

$$\frac{\partial^2}{\partial C^2}L(C, b) = \frac{\partial}{\partial C}\left(-\tilde{\Sigma}_c + C\tilde{\Sigma}_c + b\mu^\top + 2\alpha^{-2}C\right) \tag{25}$$

$$= \tilde{\Sigma}_c + 2\alpha^{-2}I \tag{26}$$

$$\frac{\partial}{\partial b}L(C, b) = \frac{\partial}{\partial b}(-\mu_c - C\mu_c + b) \tag{27}$$

$$= I \tag{28}$$

Because both $\tilde{\Sigma}_c + 2\alpha^{-2}I$ and $I$ are positive-definite, the minimization problem is strictly convex, and there exists a unique solution. To locate this unique minimum, we set the first derivative of $L(C, b)$ to zero:

$$0 = \frac{\partial}{\partial C}L(C, b) \tag{29}$$

$$= -\tilde{\Sigma}_c + C\tilde{\Sigma}_c + b\mu^\top + 2\alpha^{-2}C \tag{30}$$

$$= -\tilde{\Sigma}_c + C(\tilde{\Sigma}_c + 2\alpha^{-2}I) + b\mu^\top \tag{31}$$

$$0 = \frac{\partial}{\partial b}L(C, b) \tag{32}$$

$$= -\mu_c - C\mu_c + b \tag{33}$$

$$b = (I - C)\mu_c \tag{34}$$

We now plug Equation 34 into Equation 31, and solve for $C$:

$$0 = -\tilde{\Sigma}_c + C\tilde{\Sigma}_c + (I - C)\mu\mu^\top + 2\alpha^{-2}C \tag{35}$$

$$C = (\tilde{\Sigma}_c - \mu\mu^\top)(\tilde{\Sigma}_c - \mu\mu^\top + 2\alpha^{-2}I)^{-1} \tag{36}$$

By substituting $\Sigma_c = \tilde{\Sigma}_c - \mu\mu^\top$ (as per Equation 4) we obtain the final results:

$$C(\Sigma_c, \alpha) = \Sigma_c(\Sigma_c + 2\alpha^{-2}I)^{-1} \tag{37}$$

$$b(\Sigma_c, \alpha) = \mu - C\mu \tag{38}$$

$$\square$$

## A.2  CONCEPTORS

Conceptors are mathematical constructs that can be used for the management of neural activations (Jaeger, 2014a). A conceptor can be visualized as a structure that describes the activational pattern, or state cloud, of a set of high-dimensional activation points using an ellipsoid (see Figure 4). This

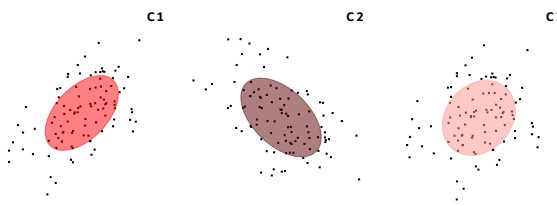

Figure 4: 2D visualization of 3 Conceptors that describe the "underlying pattern" or state space region of 3 different sets of neural activations.

conceptor is mathematically represented by a positive semi-definite matrix with eigenvalues between zero and unity that can be used to (softly) project a new set of activations toward the described ellipsoid.

Conceptors have been used to control pattern-generating RNNs effectively across various behaviors (Jaeger, 2017), prevent catastrophic forgetting and enhance continual learning in feedforward networks (He, 2023), remove bias subspaces in LLMs like BERT and GPT (Yifei et al., 2023), and distill linguistic abstractions into knowledge graphs from contextual embeddings (Kuiper, 2024; Bricman, 2022).

The eigenvalues $\mu_i$ of the conceptor matrix $C$ are defined as:

$$\mu_i = \begin{cases} \frac{\lambda_i}{\lambda_i + \alpha^{-2}} & \text{for } 0 < \lambda_i < 1 \text{ and } 0 < \alpha < \infty \\ 0 & \text{for } 0 < \lambda_i < 1 \text{ and } \alpha = 0 \\ 1 & \text{for } 0 < \lambda_i < 1 \text{ and } \alpha = \infty \\ 0 & \text{for } \lambda_i = 0 \text{ and } 0 \leq \alpha \leq \infty \\ 1 & \text{for } \lambda_i = 1 \text{ and } 0 \leq \alpha \leq \infty \end{cases}$$

where $\lambda_i$ represents the eigenvalues of the correlation matrix $R$. These eigenvalues $\mu_i$ fall within the interval $[0, 1]$ and are influenced by the aperture parameter $\alpha$. When $\alpha$ is large, the eigenvalues $\mu_i$ approach 1 and $C$ approaches the identity matrix, causing the conceptor to allow for more signal components to pass through the projection of the states with the conceptor matrix $Cx$. Conversely, when $\alpha$ is small, the eigenvalues $\mu_i$ approach 0, causing the conceptor to allow for less variability. In the extreme case of $\alpha = 0$, the conceptor collapses to the zero mapping.

### A.3 EXPERIMENTAL DETAILS

All experiments were run on NVIDIA GPUs. The GPT-NeoX model was run on one NVIDIA RTX A6000 with 48GB of VRAM, the GPT-J model was run on one NVIDIA GeForce RTX 4090 with 24GB of VRAM, and the GPT-2 Small model was run on one NVIDIA L4 Tensor Core GPU with 24GB of VRAM. Each hyperparameter sweep took less than 18 hours of compute time per model and per task.

#### A.3.1 FUNCTION STEERING

All the experimental configurations (number of experiments, number of ICL prompts and examples per prompt, accuracy metric, etc.) were, unless mentioned otherwise, adopted from Todd et al. (2024) to ensure comparability of results.

For each experiment, to generate the 4 steering mechanisms, we first compile $N_p = 100$ (ICL) prompts that demonstrate the respective input-output function. The prompts are formed by randomly sampling $N = 10$ input-output pairs from the function pairs dataset. If for a specific function, the dataset contains less than $N_p \times N = 1000$ input-output examples, this sampling is done with replacement. For each prompt $p_i^f$, the last input-output pair has the output stripped, resulting in the format:

$$p_i^f = "x_1 : y_1, x_2 : y_2, ..., x_{N-1} : y_{N-1}, x_N : "$$

where $x$ represents the input tokens of a randomly sampled (input, output) pair, $y$ represents the corresponding output tokens, $N$ represents the number of sampled input-output pairs, and $i \in \{1, \ldots, N_p\}$. A very simple example where $N_p = 3$ and $N = 3$ can be seen in Figure 5a.

old:young, vanish:appear, dark:
awake:asleep, future:past, joy: → $\bar{\mathbf{h}}_1^f$
top:bottom, tall:short, accept:

simple: $+ \,\overline{\mathbf{h}_1^f} =$ complex
encode: $+ \,\overline{\mathbf{h}_1^f} =$ decode

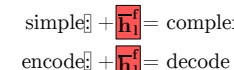

(a) Extraction of the antonym function (steering) vector $\bar{h}_\ell^f$ at layer $l$ using 3 ICL prompts.

(b) Antonym steering vector in 2 zero-shot contexts.

Figure 5: Visualization of how an antonym function (steering) vector can be extracted and applied. Example from Todd et al. (2024)

.

Formally, for each function $f \in F$ in our set of in-context learning (ICL) tasks, we have compiled a set $P_f$ of ICL prompts $p_i^f \in P_f$. Each prompt $p_i^f$ is a sequence of tokens with $N$ input-output exemplar pairs $(x, y)$ that demonstrate the function $f$ mapping between $x$ and $y$. For each experiment, we generate $N_p$ such prompts.

Now that the ICL prompts have been generated, we need to extract the relevant activations. Todd et al. (2024) showed that the neural representations of the functions are encoded in the activation vector of the last token (":") of the prompt, right before the transformer would auto-regressively start generating the output token(s). Moreover, the point in the residual stream $h$ at which the functions were most strongly encoded was shown to be at the beginning of layers $L = \{9, \ldots, 16\}$, right before MHA and FFN (Todd et al., 2024).

Formally, for each function $f \in F$ and each prompt $p_i^f \in P_f$, the activation vectors $h_\ell^f(p_i^f)$ are extracted from the residual stream $h$ at each relevant layer $l \in L$ from the last token's (":") activation vector.

For each function $f \in F$ and each layer $l \in L$, we now have $N_p$ cached activation vectors $h_\ell^f(p_i^f)$ aimed to encode the neural representation of $f$ at layer $l$. Using this, we can generate the layer-specific steering mechanisms for each function as described in Section 3.

To test the performance of the generated steering mechanisms, new sets of $N_t = 1000$ input-output pairs are randomly sampled from the function pairs dataset for each experiment. This is done with replacement for functions where the dataset contains less than $N_t$ pairs. An input prompt $p_t$ is formatted as $p_t = "x : "$, where $x$ is a tokenized input from an input-output pair. The tokenized output $y$ from the pair is left out from $p_t$ as it will be used to test the accuracy of the steering mechanisms. For each experiment, we now have $N_t$ test input prompts $p_t$.

To test the accuracy of the steering mechanisms, we apply the layer-specific steering mechanisms on independent forward passes and record their subsequent output. This means that for our experimental configuration, across the functions $f \in F$, the 5 experiments, the 4 steering mechanisms (excluding the baseline), the $N_t$ number of test prompts, and the number of layers $l \in L$, there will be $6 \times 5 \times 4 \times 1000 \times 8 = 960,000$ forward passes, each with a steering intervention.

Each steering intervention will consist of a layer-specific steering mechanism modifying the residual stream $h$ at the mechanisms' respective layer $l$. This modification can be defined as transforming the unmodified residual stream activation vector $h_\ell$ into the steered activation vector $h_\ell'$. The steering mechanisms' modification are described in Section 3.

After the respective modifications have been made to the residual stream, the forward passes will continue as usual. At the end of each forward pass, the final logits are converted into probabilities using a softmax, and the token with the highest probability is selected. This means that at the end of one experiment, we have $N_t$ single-token outputs for each layer-specific steering mechanism. These tokens can now be compared with the first token of output $y$ that corresponds with the input $x$ of the initial prompt $p_t$. Based on how many of the $N_t$ outputs were correctly identified, a top-1 accuracy is calculated for each layer-specific steering mechanism. This experiment is repeated 5 times for each function $f \in F$ to account for variability caused by the random sampling for the generation of the steering mechanisms and test sets.

### A.3.2 HYPERPARAMETER OPTIMIZATION

The performance of the steering mechanisms in the function vector experiments was optimized through a grid search over all hyperparameters. Firstly, we try steering at each layer of the model. For conceptor-based steering, we do a grid search for the aperture value $\alpha$ with possible values from $\{0.001, 0.0125, 0.05, 0.1\}$ and the scaling coefficient $\beta_c$ with possible values from $\{0.5, 1.0, 2.0, 3.0, 4.0, 5.0\}$. For additive steering, we run a grid search over the scaling coefficient $\beta_{\text{add}}$ with possible values from $\{0.5, 1.0, 1.5, 2.0, 2.5, 3.0, 4.0, 5.0\}$. The results from these hyperparameter sweeps are shown in Appendix A.5

### A.4 AFFINE CONCEPTORS AND MEAN-CENTERED VECTORS FOR FUNCTION STEERING

An important improvement for additive steering is a technique called *mean-centering*, put forward by Jorgensen et al. (2023a). This method enhances the effectiveness of steering vectors by reducing the inherent bias present in the activation space of LLMs. Activation vectors in LLMs tend to be anisotropic, meaning that they are not evenly distributed around the origin, but are instead offset in a consistent direction. This can negatively impact the steering vector's performance as the bias vector $b$ representing this offset, does not encode any specific task-related information, diluting the steering vector's effectiveness.

First, the steering vector $\bar{h}_\ell^f$ for a specific function $f$ is computed by averaging the activations at layer $\ell$ on a set of ICL prompts demonstrating the input-output function $P_f$. $\bar{h}_\ell^f$ now encodes the task-specific behavior but may still be affected by biases in the model's overall activation space. Mean-centering attempts to mitigate this by subtracting the mean activation of a broader dataset that represents the general activation space of the model. This is done by computing the mean activation vector $\mu_{\text{train}}$ over a large, representative set of prompts $D_{\text{train}}$ from the model's training data.

The mean activation vector $\mu_{\text{train}}$ was calculated using the same procedure described by Jorgensen et al. (2023a): A subset from the dataset used to train GPT-2 was compiled Gokaslan et al. (2019). The subset was constructed by storing all entries from the folders `urlsf_subset01-1/data` and `urlsf_subset01-182/data`. After this, only entries that contained less than 500 tokens (using the GPT-2 Tokenizer) were retained. This resulted in 210 entries from which the final 10 were removed, leaving a dataset of 200 entries. The mean activation vector $\mu_{\text{train}}$ was then computed by averaging the activations over this dataset.

Implementing the mean-centering performance enhancement for steering toward the execution of functions can be done as follows:

$$\bar{h}_\ell^{f,\text{mc}} = \bar{h}_\ell^f - \mu_{\text{train}} \quad \text{with} \quad \mu_{\text{train}} = \frac{1}{|D_{\text{train}}|} \sum_{d \in D_{\text{train}}} h_\ell(d) \tag{39}$$

where $\bar{h}_\ell^f$ is the activation vector at layer $\ell$, and $D_{\text{train}}$ is the dataset for which the mean-centered vector $\mu_{\text{train}}$ is computed. This refinement leads to a steering vector that can more effectively guide the model toward the specific task and has been shown to have a positive impact on the overall steering effectiveness (Jorgensen et al., 2023a).

The analogous operation of mean centering for conceptor-based steering is given by the application of affine conceptors, as derived in Section 2.4.

Table 1 in the main text and Figure 6 below show that the mean-centering mechanism provides a good improvement for both additive steering, and affine conceptors provide a (relatively smaller) improvement over linear conceptor steering. The experimental setup is as described in Appendix A.3.1.

### A.5 HYPERPARAMETER SWEEP RESULTS

In the following section, we present results from the hyperparameter optimization described in Appendix A.3.2, in order to assess the sensitivity of both steering mechanisms (additive and conceptor-based) to the hyperparameters.

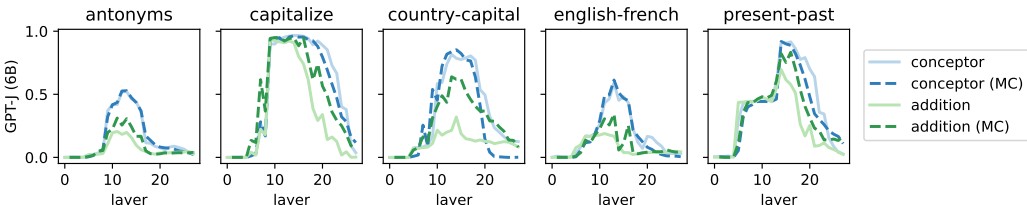

Figure 6: A comparison of additive steering, mean-centered additive steering, linear conceptor steering, and affine conceptor steering on the GPT-J (6B) model across all layers, computed on five different function vector tasks. The line shows the best average performance across five runs for the best hyperparameters for the given layer.

### A.5.1 CONCEPTOR STEERING

Figure 7 shows that the optimal choice of aperture and beta parameters for the conceptor steering mechanism is constant at $\alpha = 0.05$ and $\beta_C = 2.0$ across all tasks for the GPT-J model (for the layer with the maximum performance). Figure 8 shows similar behavior for the GPT-NeoX model, although the optimal beta parameter is $\beta = 1$ and the optimal aperture parameter changes to $\alpha = 0.0125$ for the country-capital task, and $\alpha = 0.1$ for the english-french task, and $\alpha = 0.05$ for all other tasks. This shows that hyperparameter choices are robust for conceptor steering, but still benefit from task-specific and model-specific optimization.

We further show the performance of conceptor-based steering across all layers and different beta values (taking the best-performing aperture value) for the GPT-J model in Figure 9 and for the GPT-NeoX model in Figure 10. For the GPT-J model, the best-performing layers are typically layers 12-14 with some variability (present-past being a few layers later at 14-17, and capitalize working well across layers 9-19). For the GPT-NeoX model, conceptor steering reaches (near-)maximum performance at layer 15 across all tasks, with layer 15 being at around one third of the depth of the model. Figures 11 and 12 show the performance of conceptor-based steering across all layers and different aperture values (taking the best-performing beta value) for the GPT-J model and the GPT-NeoX model, respectively, and show a similar pattern as described above.

### A.5.2 ADDITIVE STEERING

Additive steering only has two hyperparameters that were being optimized: the layer on which steering was done, and the beta value that determines the "steering strength". Figure 13 shows the performance of additive steering on the GPT-J model across all layers and beta values. Similarly to the results of conceptor-based steering, additive steering works best across layers 9-14 with peak performance always between layers 12-14. The best-performing beta values are 2.0, 3.0, and 4.0, although 2.0 is sufficient to reach peak performance for all tasks. Figure 14 shows the performance of additive steering on the GPT-NeoX model across all layers and beta values. Similar to the best-performing conceptor-based steering hyperparameters, additive steering works best on layers 12-16. The optimal beta values are 1.5 and 2.0.

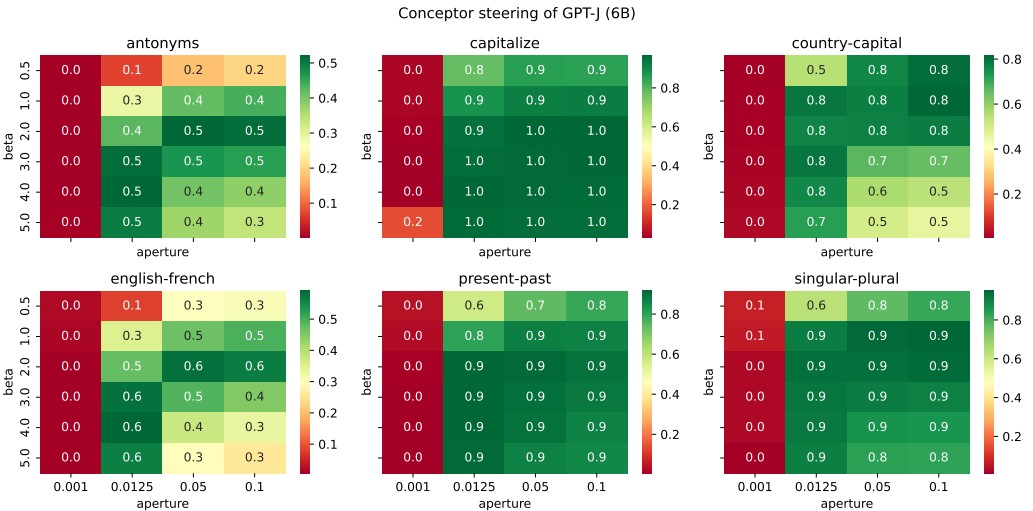

Figure 7: Performance results of the grid search across aperture and beta values (for the optimal layer) for the GPT-J (6B) model, using conceptor-based steering.

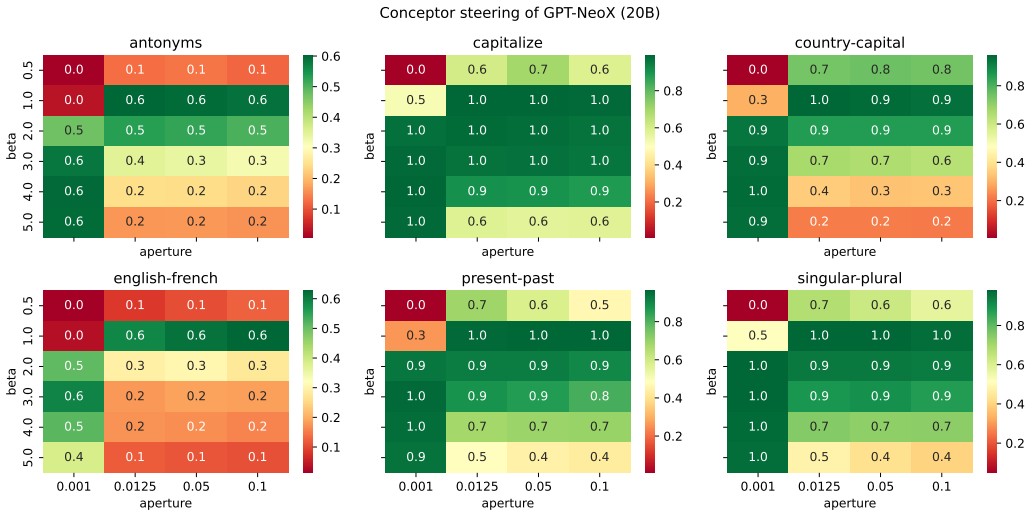

Figure 8: Performance results of the grid search across aperture and beta values (for the optimal layer) for the GPT-NeoX (20B) model, using conceptor-based steering.

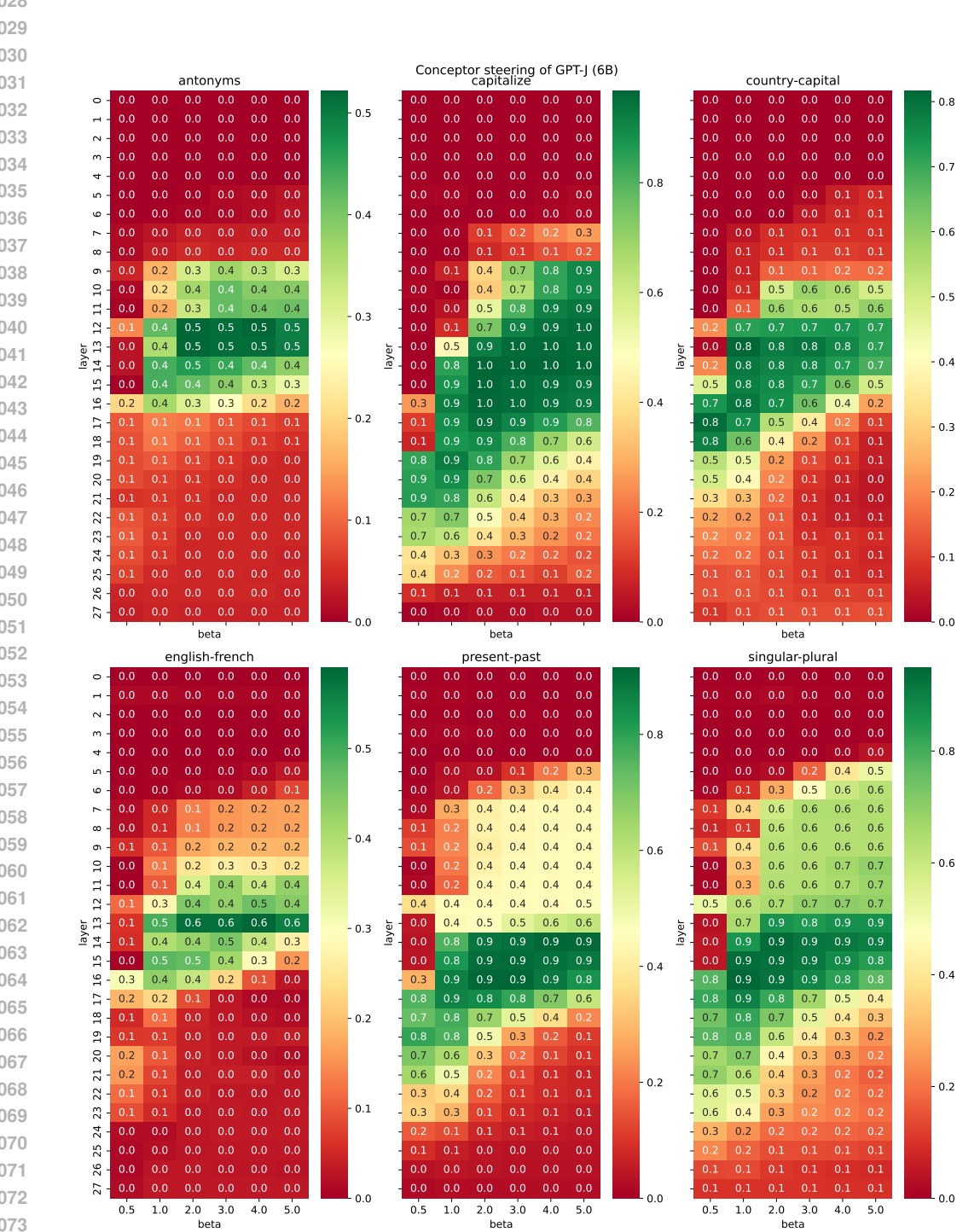

Figure 9: Performance results of the grid search across layers and beta values (for the optimal aperture value) for the GPT-J (6B) model, using conceptor-based steering.

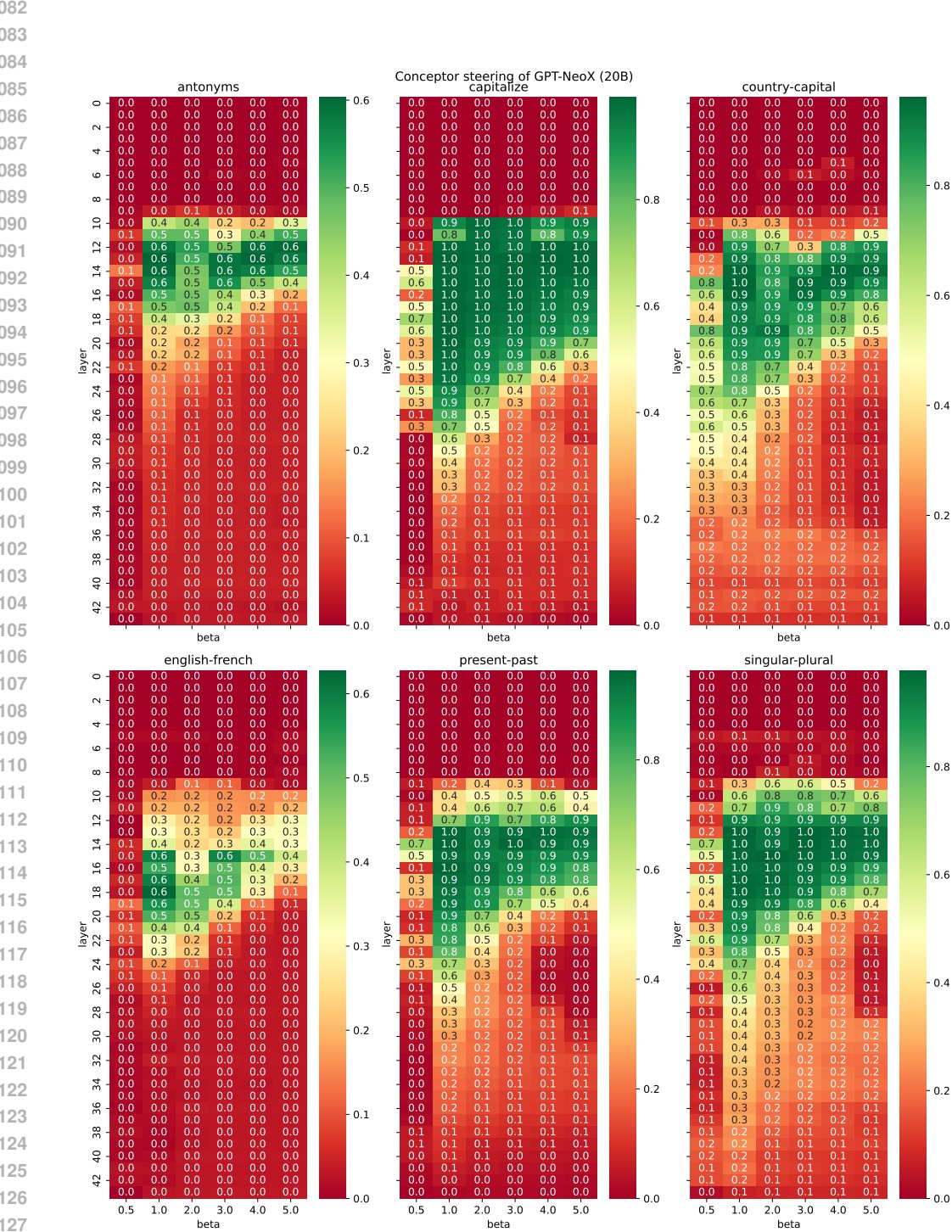

Figure 10: Performance results of the grid search across layers and beta values (for the optimal aperture value) for the GPT-NeoX (20B) model, using conceptor-based steering.

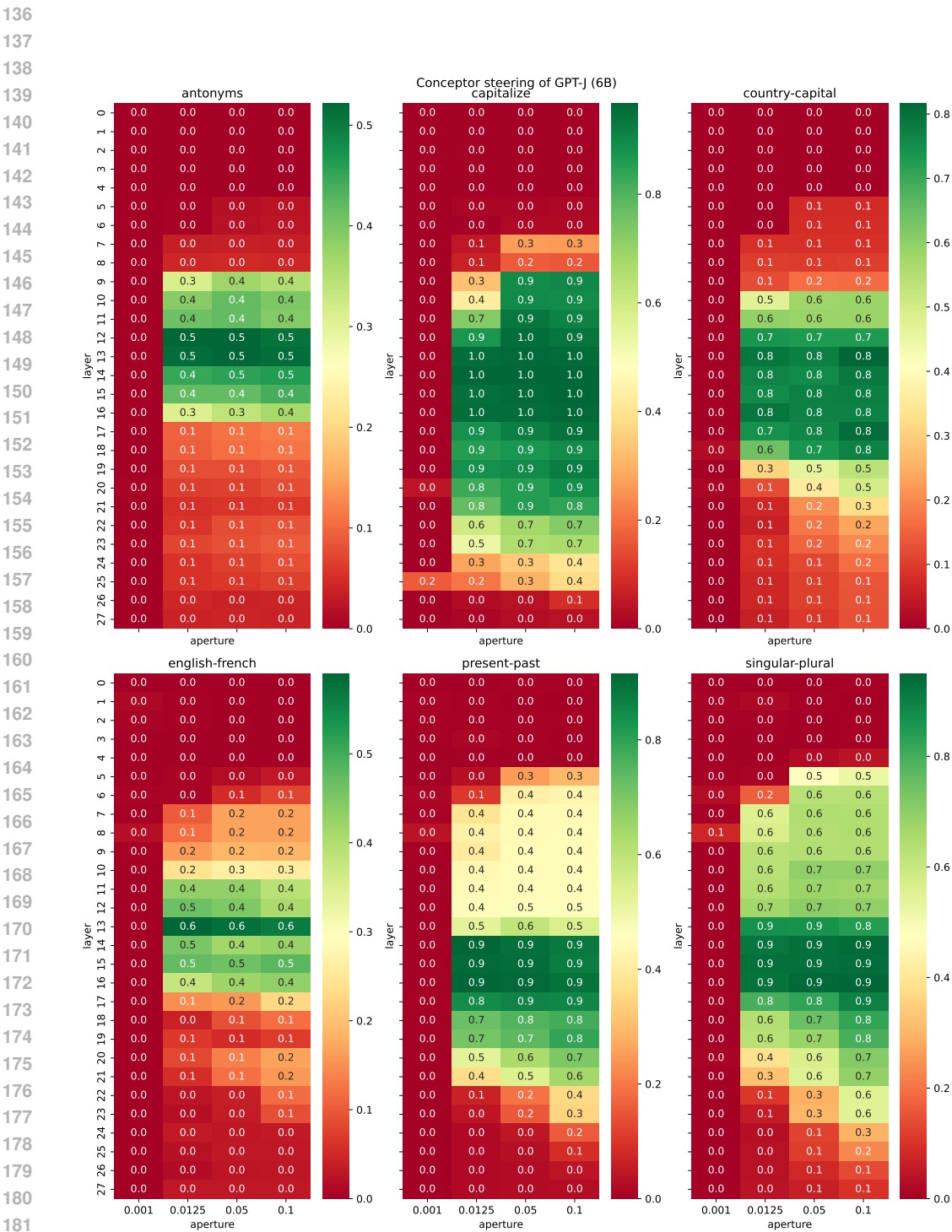

Figure 11: Performance results of the grid search across layers and aperture values (for the optimal beta value) for the GPT-J (6B) model, using conceptor-based steering.

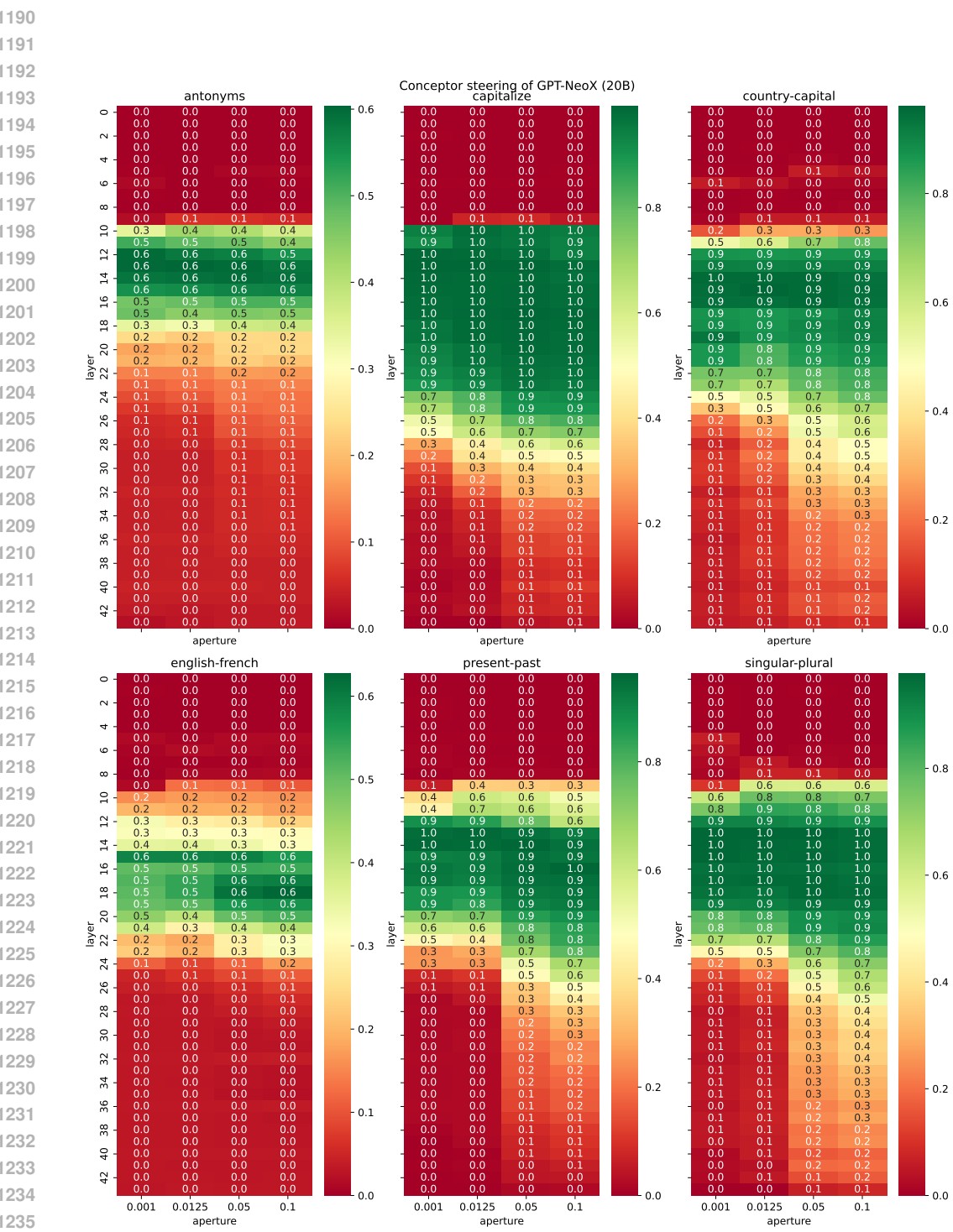

Figure 12: Performance results of the grid search across layers and aperture values (for the optimal beta value) for the GPT-NeoX (20B) model, using conceptor-based steering.

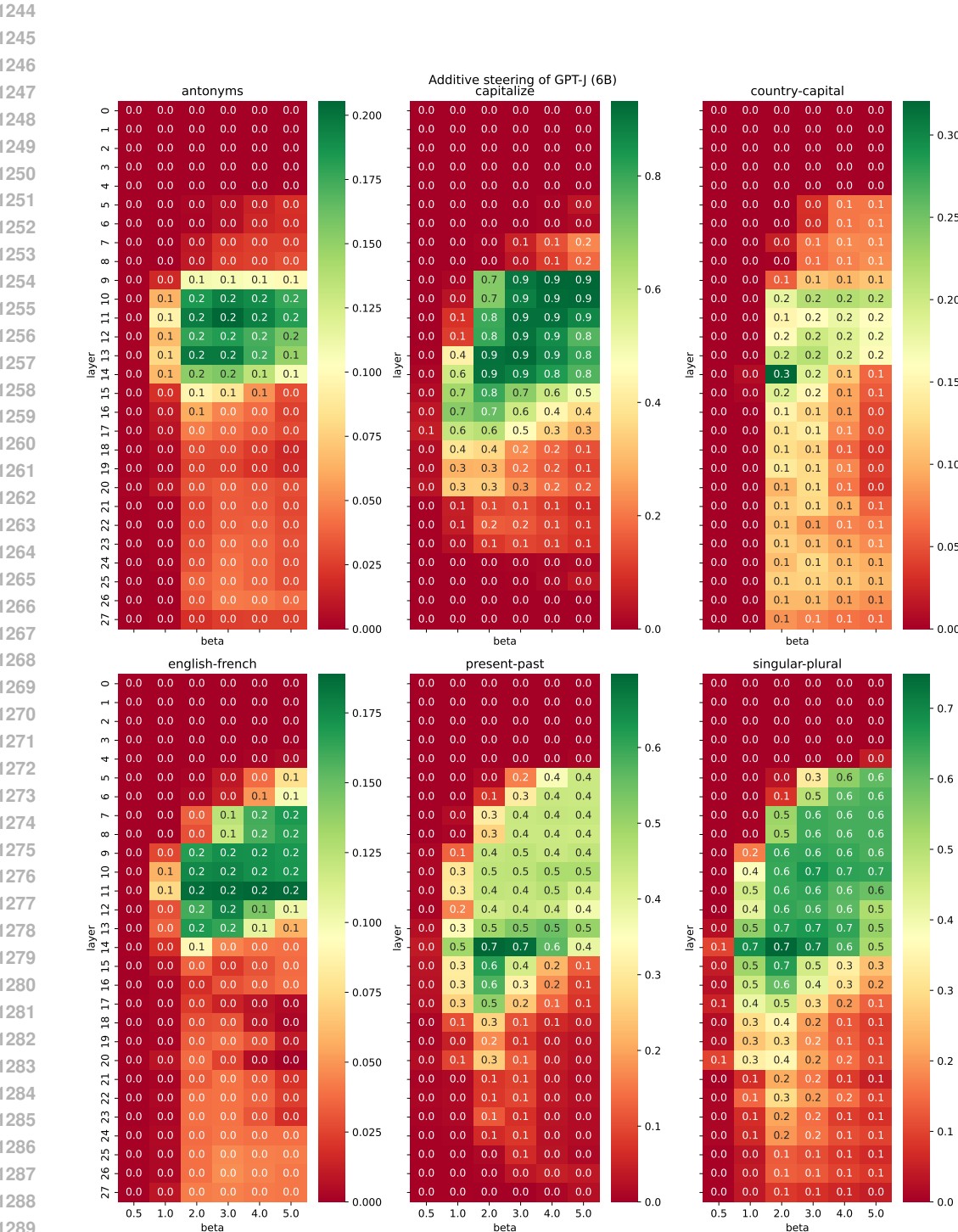

Figure 13: Performance results of the grid search across layers and beta values for the GPT-J (6B) model, using additive steering.

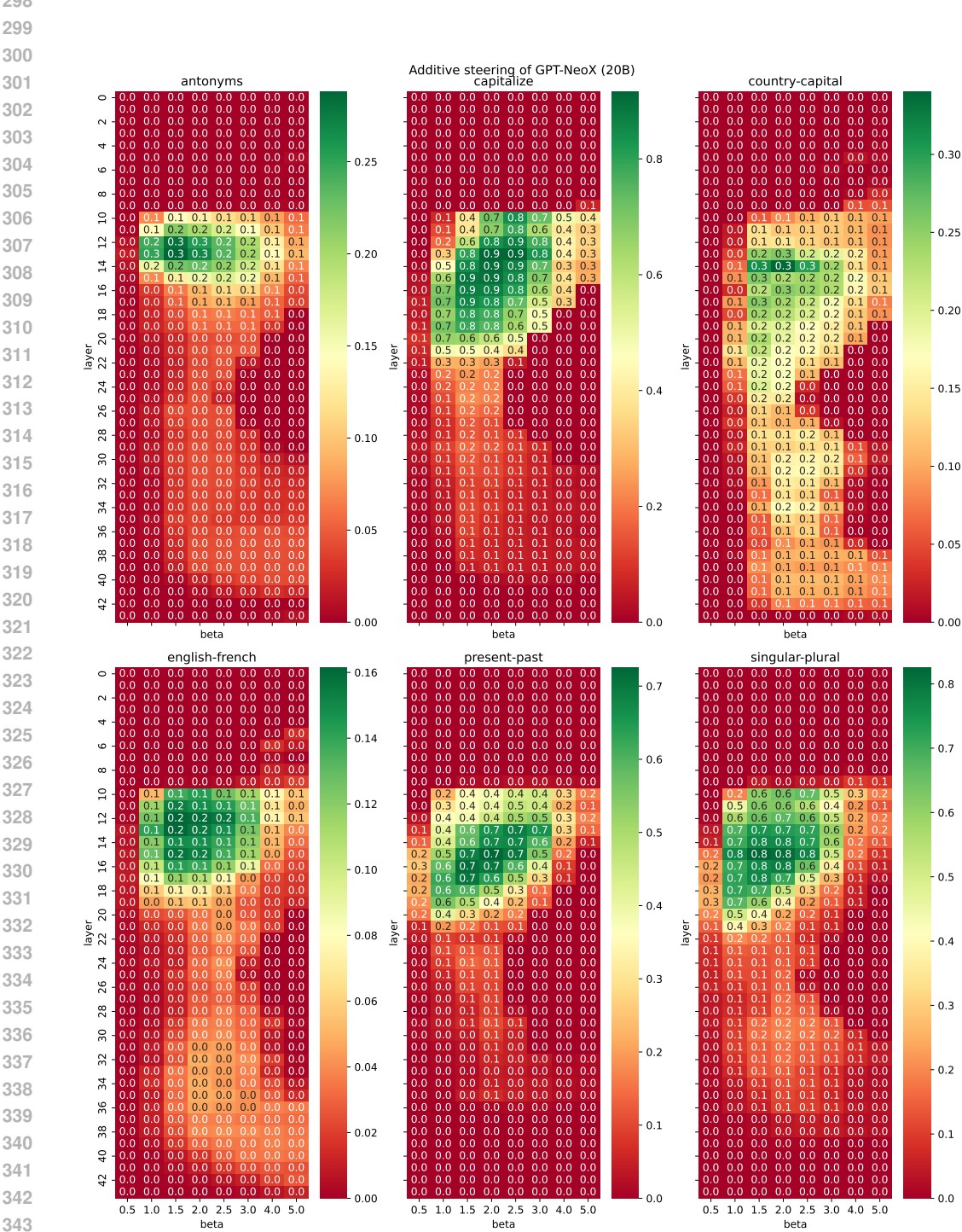

Figure 14: Performance results of the grid search across layers and beta values for the GPT-NeoX (20B) model, using additive steering.

