# OpenReview forum: "From Steering Vectors to Conceptors and Beyond: Compositional Affine Steering Mechanisms for LLMs"
_ICLR.cc/2025/Conference — ICLR 2025 Conference Withdrawn Submission_

### Official Review · Reviewer_HHvp · 2024-10-30

**Soundness:** 3
**Presentation:** 3
**Contribution:** 3
**Rating:** 6
**Confidence:** 3

**Summary:**

This paper presents a novel method for activation steering of LLMs based on Connectors. This method goes beyond the additive steering methods of previous works to use affine steering, applying an affine transformation to activations during the forward pass. This affine transformation is calculated from the matrix of activations of the concept in question. The paper presents formal and theoretical results describing conceptors and proving how to calculate the optimal affine transformation for a given concept, occasionally seeing links between this work and prior methods for activation steering. The paper then presents empirical results on 5 function steering tasks from previous work, where their method outperforms the baselines across all layers of both models. They also present results demonstrating how connectors can be combined with boolean logical operations, showing promising results.

**Strengths:**

The paper was well-written and easy to read. The contribution is interesting and novel, and shows that activation steering work can benefit from theoretical and formal thinking. While connectors are not novel, connecting them to activation steering and computing optimal connectors in this setting is, and is a worthwhile contribution. The experiments compare against reasonable baselines and show promising results. Activation steering is an exciting and important area of research, so this contribution is timely and significant in that respect.

**Weaknesses:**

# Paper being self-contained

While the paper is mostly self-contained, it references Jaeger's (2014) work on conceptors frequently. The main places where this makes the motivation and contribution less clear is in the definition from Jaeger of optimal connectors. It is not clear to me why this definition is the correct notion of optimality, and so justifying this more would be beneficial to the clarity of the paper.

# Minimal experiments

While the paper compares against several baselines, it only shows results on 5 tasks, which is quite a small set. It would be beneficial to expand the experiments in the paper in several ways:
* More and different tasks. These could be additional function vector tasks, but also the persona tasks as in https://arxiv.org/abs/2312.06681, https://arxiv.org/abs/2407.12404, or other more generative style-based tasks from previous work. This would demonstrate the benefits of this method more generally across a wider range of settings, and be much more compelling
* Measuring general performance degradation. As shown in recent work (https://www.anthropic.com/research/evaluating-feature-steering), activation steering methods can sometimes decrease general performance while increasing task-specific performance. As the conceptor method applies a more substantial transformation than additive steering, it would be beneficial to ensure this transformation doesn’t degrade general model performance more than activation steering. This could be done on generative tasks as in previous and concurrent work (https://www.anthropic.com/research/evaluating-feature-steering, https://arxiv.org/abs/2312.06681)

# Summary

Overall, I'm giving this paper a 6, as I believe the contributions of conceptors applying to activation steering and experiments are sufficiently meaningful as a contribution. To raise my score higher, I would want to see more additional experiments as described above. If experiments were done in a range of domains and results were still positive, I think this would be a very strong paper.

**Questions:**

* Why is Jaeger’s notion of optimality for a conceptor the correct one for the activation steering setting.
* How much additional computation cost does this method use over activation steering? It would be useful to have a big O notation idea of computational complexity, as if it scales quadratically with dataset size rather than linearly that is a downside of the method that should be mentioned.

---

### Official Review · Reviewer_P8qx · 2024-11-01

**Soundness:** 2
**Presentation:** 2
**Contribution:** 2
**Rating:** 5
**Confidence:** 4

**Summary:**

The authors propose a novel 'conceptor'-based class of steering functions for activation engineering. They derive a Boolean algebra for the logical composition of different conceptors. They compare their approach to previous work on function vectors and steering vectors and show that conceptors achieve a strictly higher steering accuracy for all tasks at all choices of layer.

**Strengths:**

Originality: Good. The method being proposed is novel and differs substantially from other works within the space of steering interventions.

Quality: Fair. While the principles seem sound, I have some concerns as to the validity of the experimental results, as well as the authors' claims. More details are provided in 'Weaknesses' below.

Clarity: Fair. The bulk of the paper focuses on building up to deriving the optimal linear and affine steering functions (Propositions 1,2). I found this derivation hard to follow as the derivation introduced many distinct terms (Definitions 1-4, 8-9) which were difficult to keep track of mentally. Furthermore, relatively little mental scaffolding was included to help the reader build intuition about the meaning of the intermediate steps. I would have preferred if the derivation had been summarised and / or moved to the Appendix, with the main paper simply stating the optimal affine steering intervention (Proposition 2) and focusing on building intuition for the reader. This would also have cleared up more space for results.

Significance: Fair, assuming the results and claims in the paper are correct. The paper provides a sign of life that conceptors can outperform simpler steering approaches. However, this method is also more difficult to apply, requiring an additional hyperparameter search, which may not be justified by the relative improvement over simpler, hyperparameter-free baselines. From an interpretability perspective, there is a lack of a central take-away insight from the paper. As a point of reference, the original function vectors paper (Todd et al, 2024) yields the insight that 'language models represent concepts as vectors' in activation space. The conceptor-based approach does not yield a similarly crisp insight into the geometry of language models' representation space. From an empirical alignment perspective, the results are relatively shallow (only 1 main set of experiments). Furthermore, the tasks considered are toy-ish and do not reflect realistic use cases for language model alignment. Overall, the paper is of limited significance to the broader alignment field.

**Weaknesses:**

The advantage of conceptors over baselines seems somewhat incremental. In Table 1, for 2 out of 5 tasks (capitalize and present-past), conceptors did not meaningfully outperform the addition baselines. On the remaining 3 out of 5 tasks, a substantial fraction of the improvement can be attributed to mean-centering (compare Addition-MC with Addition). Furthermore, while the paper seems to propose affine conceptors as the most general method, the benefit over linear conceptors was negligible.

I am concerned that the demonstrated advantage of conceptors could be partially due to hyperparameter optimization. Conceptor-based steering introduces an additional 'aperture' parameter which is optimised on a per-dataset basis, whereas the baseline of steering vectors does not require such a parameter. In order to do a fair comparison, I think the authors should restrict themselves to a single global choice of aperture parameter across the 6 tasks. It would also be important to include a discussion on the effect of non-optimal choices of the aperture parameter.

The paper claims to unify affine steering functions (Singh et al, 2024) and additive steering functions (Turner et al, 2023). However, the experiments section only includes comparisons to the latter. Given the scope of the authors' claims, it seems important to also have the comparisons to the former.

Little analysis and discussion is provided to allow the reader to understand why conceptors improve over baselines. I think the paper would be much more exciting if it could state the specific assumptions on representational geometry that conceptors target (similar to how steering vectors target linear geometry), and then demonstrate that this geometrical structure is present in language models. See [1] for a reference work where I think this is achieved.

As stated above, the methods section is difficult to follow and could probably be shortened.

[1] https://arxiv.org/abs/2311.03658

**Questions:**

What is the purpose of Definition 1? It seems like phi-assisted steering functions are not referred to again after being introduced here.

What is the motivation for solving the specific optimization problem in Definition 4? How do the individual terms in the loss function connect to downstream properties we care about? In particular, I’m not sure why we want the steering function C to satisfy H_c \approx CH_c, which is what the first term is incentivising, and I’m also not sure why we want C to have small norm, which is what the second term is incentivising.

In Figure 1, the conceptor steering operation is illustrated as 'projecting' all activations to within an ellipsoid space. Why does this make sense to do (in terms of the language model's representational geometry)?

---

### Official Review · Reviewer_FswL · 2024-11-03

**Soundness:** 4
**Presentation:** 3
**Contribution:** 3
**Rating:** 6
**Confidence:** 4

**Summary:**

The paper explores the theory and practice of conceptors, a tool developed for controlling recurrent neural networks, to steer the output of a Large Language Model. It first generalizes the theory of conceptors, highlighting how the optimal steering functions can be computed by estimating statistics of an LLM's activations, and then evaluates the method on standard activation steering benchmarks, showing it outperforms common approaches.

**Strengths:**

- The approach provides a theoretically motivated alternative to existing (e.g., contrastive) LLM activation steering methods.
- The method outperforms the standard additive activation steering approaches on commonly employed benchmarks.
- The paper nicely connects previous work on recurrent neural networks and modern efforts in controlling Large Language Models.

**Weaknesses:**

- No error bars are reported in any of the plots and table.
- There are some minor formatting issues in the paper (e.g., see the positioning of Figure 3).
- Only rather outdated and small open weights models are used for the analysis.
- There are potential concerns about computational cost compared to alternatives.

**Questions:**

- What are the limitations of the approach in terms of computational cost? Does it get prohibitive when larger models (e.g., the largest available open weight models) are used?
- What is the variation across runs? Can you plot the error bars?
- Can you give a more complete explanation of why it is justified to call the steering function defined in Definition 9 as optimal?
- Could you evaluate the method with a more recent LLM? (e.g., Llama)

---

### Official Review · Reviewer_5mdn · 2024-11-04

**Soundness:** 2
**Presentation:** 1
**Contribution:** 2
**Rating:** 3
**Confidence:** 3

**Summary:**

This paper introduces a new method for affine steering in LLMs by using conceptors. The author uses conceptor theory to present a theoretical framework for activation steering and conducts experiment to demonstrate the effectiveness of the proposed method.

**Strengths:**

1. The author empirically shows the effectiveness of conceptor steering.

**Weaknesses:**

1. Figure 1 is not referenced in the main text and lacks description (what are green and yellow dots and I'm still confused what's the difference between additive and conceptor steering).
2. Many terms not well-explained (see questions below) and lacks clarity, making this paper hard to understand.
3. The use of "performant" is abrupt in line 210. How to define "performant" and what makes previous method not "performant"?
4. Section 3 is unclear. The method used here can be better explained.

Minor issues:
1. Line 184 "$\rightarrow$" instead of "$\mapsto$".
2. Line 230 $h$ --> $\mathbf{H}$?

**Questions:**

1. What is $\phi(s)$ in line 144?
2. What is $c'$ in equation (5)? Is it equivalent as saying $\phi(s)\neq c$ for the first line?
3. What does "optimal" mean in line 203? What is guardedness in line 202 and 208?
4. "As we do not want to rely on the concept function $\phi$ to apply our steering function $\phi$ to apply our steering function, we instead rely only on the concept-conditional convariance matrix $\Sigma_c$ -- why? The motivation is not well-explained.
5. For Figure 2 and 3 can you show the accuracy when provided with in-context examples?
6. Overall I am confused on how the proposed method works explicitly and how does it different from the procedure in Todd et al. (2024)? I would like to understand the difference so I can better understand the proposed method in this paper. Furthermore, I was a bit confused what does a better steering method mean. By looking at plot 2 and 3, can we say that a steering method is better if it can better extract features to represent the task?

---
Overall by looking at the experiment result it seems this paper proposes a method that extracts task features more effectively but the paper lacks clarity on how this method works.

---

### Note · Authors · 2024-12-02

I have read and agree with the venue's withdrawal policy on behalf of myself and my co-authors.